# SoundNet: Learning Sound Representations from Unlabeled Video

**Yusuf Aytar**\*
MIT
yusuf@csail.mit.edu

**Carl Vondrick**\*
MIT
vondrick@mit.edu

**Antonio Torralba**
MIT
torralba@mit.edu

## Abstract

We learn rich natural sound representations by capitalizing on large amounts of unlabeled sound data collected in the wild. We leverage the natural synchronization between vision and sound to learn an acoustic representation using two-million unlabeled videos. Unlabeled video has the advantage that it can be economically acquired at massive scales, yet contains useful signals about natural sound. We propose a student-teacher training procedure which transfers discriminative visual knowledge from well established visual recognition models into the sound modality using unlabeled video as a bridge. Our sound representation yields significant performance improvements over the state-of-the-art results on standard benchmarks for acoustic scene/object classification. Visualizations suggest some high-level semantics automatically emerge in the sound network, even though it is trained without ground truth labels.

## 1   Introduction

The fields of object recognition, speech recognition, machine translation have been revolutionized by the emergence of massive labeled datasets [31, 42, 10] and learned deep representations [17, 33, 10, 35]. However, there has not yet been the same corresponding progress in natural sound understanding tasks. We attribute this partly to the lack of large labeled datasets of sound, which are often both expensive and ambiguous to collect. We believe that large-scale sound data can also significantly advance natural sound understanding. In this paper, we leverage over one year of sounds collected in-the-wild to learn semantically rich sound representations.

We propose to scale up by capitalizing on the natural synchronization between vision and sound to learn an acoustic representation from unlabeled video. Unlabeled video has the advantage that it can be economically acquired at massive scales, yet contains useful signals about sound. Recent progress in computer vision has enabled machines to recognize scenes and objects in images and videos with good accuracy. We show how to transfer this discriminative visual knowledge into sound using unlabeled video as a bridge.

We present a deep convolutional network that learns directly on raw audio waveforms, which is trained by transferring knowledge from vision into sound. Although the network is trained with visual supervision, the network has no dependence on vision during inference. In our experiments, we show that the representation learned by our network obtains state-of-the-art accuracy on three standard acoustic scene classification datasets. Since we can leverage large amounts of unlabeled sound data, it is feasible to train deeper networks without significant overfitting, and our experiments suggest deeper models perform better. Visualizations of the representation suggest that the network is also learning high-level detectors, such as recognizing bird chirps or crowds cheering, even though it is trained directly from audio without ground truth labels.

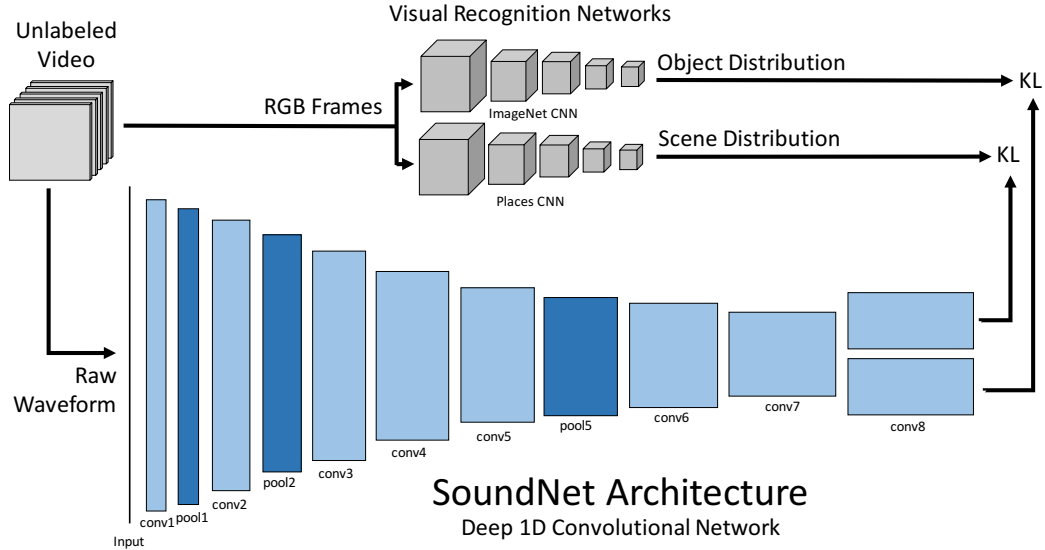

Figure 1: **SoundNet:** We propose a deep convolutional architecture for natural sound recognition. We train the network by transferring discriminative knowledge from visual recognition networks into sound networks. Our approach capitalizes on the synchronization of vision and sound in video.

The primary contribution of this paper is the development of a large-scale and semantically rich representation for natural sound. We believe large-scale models of natural sounds can have a large impact in many real-world applications, such as robotics and cross-modal understanding. The remainder of this paper describes our method and experiments in detail. We first review related work. In section 2, we describe our unlabeled video dataset and in section 3 we present our network and training procedure. Finally in section 4 we conclude with experiments on standard benchmarks and show several visualizations of the learned representation. Code, data, and models will be released.

## 1.1 Related Work

**Sound Recognition:** Although large-scale audio understanding has been extensively studied in the context of music [5, 37] and speech recognition [10], we focus on understanding natural, in-the-wild sounds. Acoustic scene classification, classifying sound excerpts into existing acoustic scene/object categories, is predominantly based on applying a variety of general classifiers (SVMs, GMMs, etc.) to the manually crafted sound features (MFCC, spectrograms, etc.) [4, 29, 21, 30, 34, 32]. Even though there are unsupervised [20] and supervised [27, 23, 6, 12] deep learning methods applied to sound classification, the models are limited by the amount of available labeled natural sound data. We distinguish ourselves from the existing literature by training a deep fully convolutional network on a large scale dataset (2M videos). This allows us to train much deeper networks. Another key advantage of our approach is that we supervise our sound recognition network through semantically rich visual discriminative models [33, 17] which proved their robustness on a variety of large scale object/scene categorization challenges[31, 42]. [26] also investigates the relation between vision and sound modalities, but focuses on producing sound from image sequences. Concurrent work [11] also explores video as a form of weak labeling for audio event classification.

**Transfer Learning:** Transfer learning is widely studied within computer vision such as transferring knowledge for object detection [1, 2] and segmentation [18], however transferring from vision to other modalities are only possible recently with the emergence of high performance visual models [33, 17]. Our method builds upon teacher-student models [3, 9] and dark knowledge transfer [13]. In [3, 13] the basic idea is to compress (i.e. transfer) discriminative knowledge from a well-trained complex model to a simpler model without loosing considerable accuracy. In [3] and [13] both the teacher and the student are in the same modality, whereas in our approach the teacher operates on vision to train the student model in sound. [9] also transfer visual supervision into depth models.

**Cross-Modal Learning and Unlabeled Video:** Our approach is broadly inspired by efforts to model cross-modal relations [24, 14, 7, 26] and works that leverage large amounts of unlabeled video [25, 41, 8, 40, 39]. In this work, we leverage the natural synchronization between vision and sound to learn a deep representation of natural sounds without ground truth sound labels.

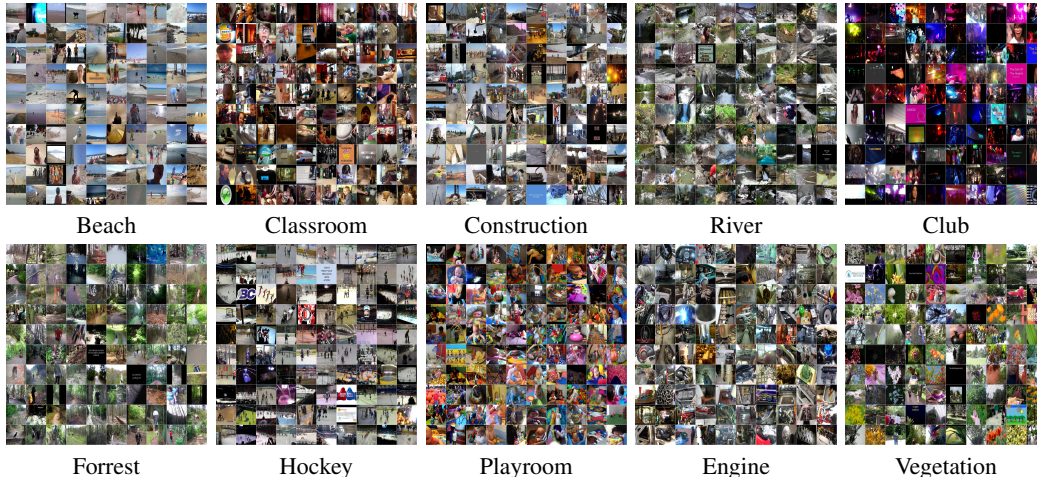

| Beach | Classroom | Construction | River | Club |
| Forrest | Hockey | Playroom | Engine | Vegetation |

Figure 2: **Unlabeled Video Dataset:** Sample frames from our 2+ million video dataset. For visualization purposes, each frame is automatically categorized by object and scene vision networks.

## 2 Large Unlabeled Video Dataset

We seek to learn a representation for sound by leveraging massive amounts of unlabeled videos. While there are a variety of sources available on the web (e.g., YouTube, Flickr), we chose to use videos from Flickr because they are natural, not professionally edited, short clips that capture various sounds in everyday, in-the-wild situations. We downloaded over two million videos from Flickr by querying for popular tags [36] and dictionary words, which resulted in over one year of continuous natural sound and video, which we use for training. The length of each video varies from a few seconds to several minutes. We show a small sample of frames from the video dataset in Figure 2.

We wish to process sound waves in the raw. Hence, the only post-processing we did on the videos was to convert sound to MP3s, reduce the sampling rate to 22 kHz, and convert to single channel audio. Although this slightly degrades the quality of the sound, it allows us to more efficiently operate on large datasets. We also scaled the waveform to be in the range $[-256, 256]$. We did not need to subtract the mean because it was naturally near zero already.

## 3 Learning Sound Representations

### 3.1 Deep Convolutional Sound Network

**Convolutional Network:** We present a deep convolutional architecture for learning sound representations. We propose to use a series of one-dimensional convolutions followed by nonlinearities (i.e. ReLU layer) in order to process sound. Convolutional networks are well-suited for audio signals for a couple of reasons. Firstly, like images [19], we desire our network to be invariant to translations, a property that reduces the number of parameters we need to learn and increases efficiency. Secondly, convolutional networks allow us to stack layers, which enables us to detect higher-level concepts through a series of lower-level detectors.

**Variable Length Input/Output:** Since sound can vary in temporal length, we desire our network to handle variable-length inputs. To do this, we use a fully convolutional network. As convolutional layers are invariant to location, we can convolve each layer depending on the length of the input. Consequently, in our architecture, we only use convolutional and pooling layers. Since the representation adapts to the input length, we must design the output layers to work with variable length inputs as well. While we could have used a global pooling strategy [37] to down-sample variable length inputs to a fixed dimensional vector, such a strategy may unnecessarily discard information useful for high-level representations. Since we ultimately aim to train this network with video, which is also variable length, we instead use a convolutional output layer to produce an output over multiple timesteps in video. This strategy is similar to a spatial loss in images [22], but instead temporally.

**Network Depth:** Since we will use a large amount of video to train, it is feasible to use deep architectures without significant over-fitting. We experiment with both five-layer and eight-layer networks.

| Layer | conv1 | pool1 | conv2 | pool2 | conv3 | conv4 | conv5 | pool5 | conv6 | conv7 | conv8 |
|---|---|---|---|---|---|---|---|---|---|---|---|
| Dim. | 220,050 | 27,506 | 13,782 | 1,722 | 862 | 432 | 217 | 54 | 28 | 15 | 4 |
| # of Filters | 16 | 16 | 32 | 32 | 64 | 128 | 256 | 256 | 512 | 1024 | 1401 |
| Filter Size | 64 | 8 | 32 | 8 | 16 | 8 | 4 | 4 | 4 | 4 | 8 |
| Stride | 2 | 1 | 2 | 1 | 2 | 2 | 2 | 1 | 2 | 2 | 2 |
| Padding | 32 | 0 | 16 | 0 | 8 | 4 | 2 | 0 | 2 | 2 | 0 |

Table 1: **SoundNet (8 Layer):** The configuration of the layers for the 8-layer SoundNet.

| conv1 | pool1 | conv2 | pool2 | conv3 | pool3 | conv4 | conv5 |
|---|---|---|---|---|---|---|---|
| 220,050 | 27,506 | 13,782 | 1,722 | 862 | 432 | 217 | 54 |
| 32 | 32 | 64 | 64 | 128 | 128 | 256 | 1401 |
| 64 | 8 | 32 | 8 | 16 | 8 | 8 | 16 |
| 2 | 8 | 2 | 8 | 2 | 8 | 2 | 12 |
| 32 | 0 | 16 | 0 | 8 | 0 | 4 | 4 |

Table 2: **SoundNet (5 Layer):** The configuration for the 5-layer SoundNet.

We visualize the eight-layer network architecture in Figure 1, which conists of $8$ convolutional layers and $3$ max-pooling layers. We show the layer configuration in Table 1 and Table 2.

### 3.2 Visual Transfer into Sound

The main idea in this paper is to leverage the natural synchronization between vision and sound in unlabeled video in order to learn a representation for sound. We model the learning problem from a student-teacher perspective. In our case, state-of-the-art networks for vision will teach our network for sound to recognize scenes and objects.

Let $x_i \in \mathbb{R}^D$ be a waveform and $y_i \in \mathbb{R}^{3 \times T \times W \times H}$ be its corresponding video for $1 \leq i \leq N$, where $W, H, T$ are width, height and number of sampled frames in the video, respectively. During learning, we aim to use the posterior probabilities from a teacher vision network $g_k(y_i)$ in order to train our student network $f_k(x_i)$ to recognize concepts given sound. As we wish to transfer knowledge from both object and scene networks, $k$ enumerates the concepts we are transferring. During learning, we optimize $\min_\theta \sum_{k=1}^{K} \sum_{i=1}^{N} D_{\mathrm{KL}} \left( g_k(y_i) || f_k(x_i; \theta) \right)$ where $D_{KL}(P||Q) = \sum_j P_j \log \frac{P_j}{Q_j}$ is the KL-divergence. While there are a variety of distance metrics we could have use, we chose KL-divergence because the outputs from the vision network $g_k$ can be interpreted as a distribution of categories. As KL-divergence is differentiable, we optimize it using back-propagation [19] and stochastic gradient descent. We transfer from both scene and object visual networks ($K = 2$).

### 3.3 Sound Classification

Although we train SoundNet to classify visual categories, the categories we wish to recognize may not appear in visual models (e.g., sneezing). Consequently, we use a different strategy to attach semantic meaning to sounds. We ignore the output layer of our network and use the internal representation as features for training classifiers, using a small amount of labeled sound data for the concepts of interest. We pick a layer in the network to use as features and train a linear SVM. For multi-class classification, we use a one-vs-all strategy. We perform cross-validation to pick the margin regularization hyper-parameter. For robustness, we follow a standard data augmentation procedure where each training sample is split into overlapping fixed length sound excerpts, which we compute features on and use for training. During inference, we average predictions across all windows.

### 3.4 Implementation

Our approach is implemented in Torch7. We use the Adam [16] optimizer and a fixed learning rate of $0.001$ and momentum term of $0.9$ throughout our experiments. We experimented with several batch sizes, and found $64$ to produce good results. We initialized all the weights to zero mean Gaussian noise with a standard deviation of $0.01$. After every convolution, we use batch normalization [15] and rectified linear activation units [17]. We train the network for $100,000$ iterations. Optimization typically took 1 day on a GPU.

## 4 Experiments

**Experimental Setup:** We split the unlabeled video dataset into a training set and a held-out validation set. We use $2,000,000$ videos for training, and the remaining $140,000$ videos for validation. After training the network, we use the hidden representation as a feature extractor for learning on smaller,

| Method | Accuracy |
|---|---|
| RG [29] | 69% |
| LTT [21] | 72% |
| RNH [30] | 77% |
| Ensemble [34] | 78% |
| **SoundNet** | **88%** |

Table 3: **Acoustic Scene Classification on DCASE:** We evaluate classification accuracy on the DCASE dataset. By leveraging large amounts of unlabeled video, SoundNet generally outperforms hand-crafted features by 10%.

| Method | Accuracy on ESC-50 | ESC-10 |
|---|---|---|
| SVM-MFCC [28] | 39.6% | 67.5% |
| Convolutional Autoencoder | 39.9% | 74.3% |
| Random Forest [28] | 44.3% | 72.7% |
| Piczak ConvNet [27] | 64.5% | 81.0% |
| **SoundNet** | **74.2%** | **92.2%** |
| Human Performance [28] | 81.3% | 95.7% |

Table 4: **Acoustic Scene Classification on ESC-50 and ESC-10:** We evaluate classification accuracy on the ESC datasets. Results suggest that deep convolutional sound networks trained with visual supervision on unlabeled data outperforms baselines.

labeled sound only datasets. We extract features for a given layer, and train an SVM on the task of interest. For training the SVM, we use the standard training/test splits of the datasets. We report classification accuracy.

**Baselines::** In addition to published baselines on standard datasets, we explored an additional baseline trained on our unlabeled videos. We experimented using a convolutional autoencoder for sound, trained over our video dataset. We use an autoencoder with 4 encoder layers and 4 decoder layers. For the encoder layers, we used the same first four convolutional layers as SoundNet. For the decoders, we used a fractionally strided convolutional layers (in order to upsample instead of downsample). Note that we experimented with deeper autoencoders, but they performed worse. We used mean squared error for the reconstruction loss, and trained the autoencoders for several days.

## 4.1 Acoustic Scene Classification

We evaluate the SoundNet representation for acoustic scene classification. The aim in this task is to categorize sound clips into one of the many acoustic scene categories. We use three standard, publicly available datasets: DCASE Challenge[34], ESC-50 [28], and ESC-10 [28].

**DCASE[34]:** One of the tasks in the Detection and Classification of Acoustic Scenes and Events Challenge (DCASE)[34] is to recognize scenes from natural sounds. In the challenge, there are 10 acoustic scene categories, 10 training examples per category, and 100 held-out testing examples. Each example is a 30 seconds audio recording. The task is to categorize natural sounds into existing 10 acoustic scene categories. Multi-class classification accuracy is used as the performance metric.

**ESC-50 and ESC-10 [28]:** The ESC-50 dataset is a collection of 2000 short (5 seconds) environmental sound recordings of equally balanced 50 categories selected from 5 major groups (animals, natural soundscapes, human non-speech sounds, interior/domestic sounds, and exterior/urban noises). Each category has 40 samples. The data is prearranged into 5 folds and the accuracy results are reported as the mean of 5 leave-one-fold-out evaluations. The performance of untrained human participants on this dataset is 81.3% [28]. ESC-10 is a subset of ESC-50 which consists of 10 classes (dog bark, rain, sea waves, baby cry, clock tic, person sneeze, helicopter, chainsaw, rooster, and fire cracking). The human performance on this dataset is 95.7%.

We have two major evaluations on this section: (a) comparison with the existing state of the art results, (b) diagnostic performance evaluation of inner layers of SoundNet as generic features for this task.

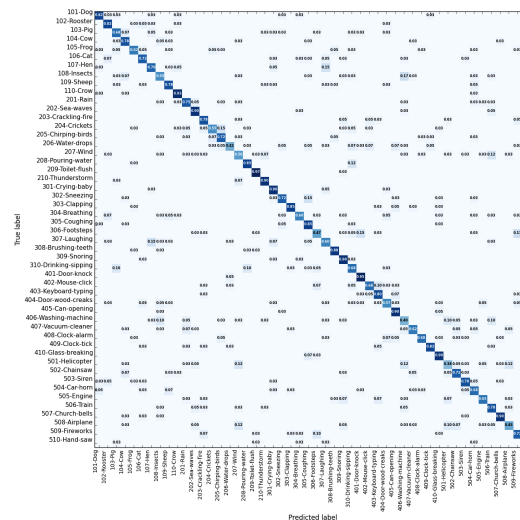

Figure 3: SoundNet confusions on ESC-50

In DCASE we used 5 second excerpts, and in ESC datasets we used 1 second windows. In both evaluations a multi-class SVM (multiple one-vs all classifiers) is trained over extracted

| Comparison of | SoundNet Model | Accuracy on | |
|---|---|---|---|
| | | ESC-50 | ESC-10 |
| Loss | 8 Layer, $\ell_2$ Loss | 47.8% | 81.5% |
| | 8 Layer, KL Loss | **72.9%** | **92.2%** |
| Teacher Net | 8 Layer, ImageNet Only | 69.5% | 89.8% |
| | 8 Layer, Places Only | 71.1% | 89.5% |
| | 8 Layer, Both | **72.9%** | **92.2%** |
| Depth and Visual Transfer | 5 Layer, Scratch Init | 65.0% | 82.3% |
| | 8 Layer, Scratch Init | 51.1% | 75.5% |
| | 5 Layer, Unlabeled Video | 66.1% | 86.8% |
| | 8 Layer, Unlabeled Video | **72.9%** | **92.2%** |

Table 5: **Ablation Analysis:** We breakdown accuracy of various configurations using `pool5` from SoundNet trained with VGG. Results suggest that deeper convolutional sound networks trained with visual supervision on unlabeled data helps recognition.

| Dataset | Model | conv4 | conv5 | pool5 | conv6 | conv7 | conv8 |
|---|---|---|---|---|---|---|---|
| DCASE [34] | 8 Layer, AlexNet | 84% | 85% | 84% | 83% | 78% | 68% |
| | 8 Layer, VGG | 77% | 88% | 88% | 87% | 84% | 74% |
| ESC50 [28] | 8 Layer, AlexNet | 66.0% | 71.2% | 74.2% | 74% | 63.8% | 45.7% |
| | 8 Layer, VGG | 66.0% | 69.3% | 72.9% | 73.3% | 59.8% | 43.7% |

Table 6: **Which layer and teacher network gives better features?** The performance comparison of extracting features at different SoundNet layers on acoustic scene/object classification tasks.

SoundNet features. Same data augmentation procedure is also applied during testing and the mean score of all sound excerpts is used as the final score of a test recording for any particular category.

**Comparison to State-of-the-Art:** Table 3 and 4 compare recognition performance of SoundNet features versus previous state-of-the-art features on three datasets. In all cases SoundNet features outperformed the existing results by around $10\%$. Interestingly, SoundNet features approach human performance on ESC-10 dataset, however we stress that this dataset may be easy. We report the confusion matrix across all folds on ESC-50 in Figure 3. The results suggest our approach obtains very good performance on categories such as toilet flush (97% accuracy) or door knocks (95% accuracy). Common confusions are laughing confused as hens, foot steps confused as door knocks, and insects confused as washing machines.

## 4.2 Ablation Analysis

To better understand our approach, we perform an ablation analysis in Table 5 and Table 6.

**Comparison of Loss and Teacher Net (Table 5):** We tried training with different subsets of target categories. In general, performance generally improves with increasing visual supervision. As expected, our results suggest that using both ImageNet and Places networks as supervision performs better than a single one. This indicates that progress in sound understanding may be furthered by building stronger vision models. We also experimented with using $\ell_2$ loss on the target outputs instead of $KL$ loss, which performed significantly worse.

**Comparison of Network Depth (Table 5):** We quantified the impact of network depth. We use five layer version of SoundNet (instead of the full eight) as a feature extractor instead. The five-layer SoundNet architecture performed 8% worse than the eight-layer architecture, suggesting depth is helpful for sound understanding. Interestingly, the five-layer network still generally outperforms previous state-of-the-art baselines, but the margin is less. We hypothesize even deeper networks may perform better, which can be trained without significant over-fitting by leveraging large amounts of unlabeled video.

**Comparison of Supervision (Table 5):** We also experimented with training the network without video by using only the labeled target training set, which is relatively small (thousands of examples). We simply change the network to output the class probabilities, and train it from random initialization with a cross entropy loss. Hence, the only change is that this baseline does not use any unlabeled video, allowing us to quantify the contribution of unlabeled video. The five layer SoundNet achieves slightly better results than [27] which is also a convolutional network trained with same data but with a different architecture, suggesting our five layer architecture is similar. Increasing the depth from five layers to eight layers decreases the performance from $65\%$ to $51\%$, probably because it overfits to the small training set. However, when trained with visual transfer from unlabeled video, the eight layer SoundNet achieves a significant gain of around $20\%$ compared to the five layer version. This

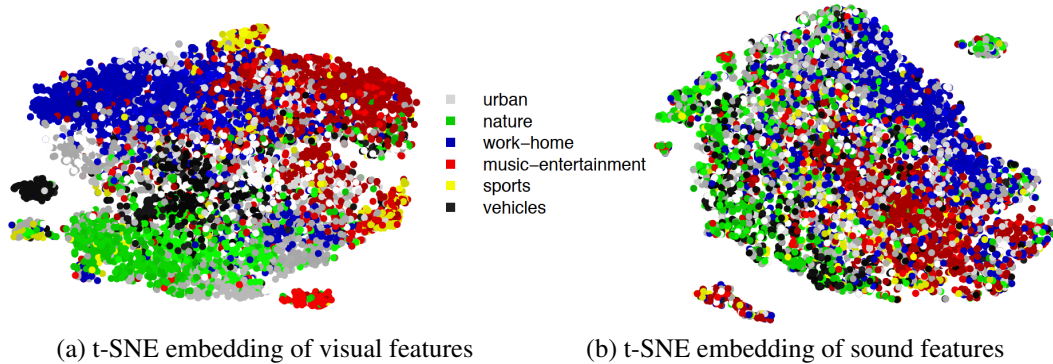

|  | urban |
|  | nature |
|  | work–home |
|  | music–entertainment |
|  | sports |
|  | vehicles |

(a) t-SNE embedding of visual features    (b) t-SNE embedding of sound features

Figure 4: **t-SNE embeddings** using visual features and sound features (SoundNet `conv7`). The visual features are concatenated `fc7` features from the VGG networks for ImageNet and Places2. Note that t-SNE embeddings do not use the class labels. Labels are only used during final visualization.

| Feature | sound | vision | vision+sound |
|---|---|---|---|
| 8 Layer, conv7 | 32.4% | 49.4% | 51.4% |
| 8 Layer, conv8 | 32.3% | 49.4% | 50.5% |

Table 7: **Multi-Modal Recognition:** We report classification accuracy on $\sim 4K$ labeled test videos over 44 categories.

suggests that unlabeled video is a powerful signal for sound understanding, and it can be acquired at large enough scales to support training high-capacity deep networks.

**Comparison of Layer and Teacher Network (Table 6):** We analyze the discriminative performance of each SoundNet layer. Generally, features from the `pool5` layer gives the best performance. We also compared different teacher networks for visual supervision (either VGGNet or AlexNet). The results are inconclusive on which teacher network to use: VGG is a better teacher network for DCASE while AlexNet is a better teacher network for ESC50.

### 4.3 Multi-Modal Recognition

In order to compare sound features with visual features on scene/object categorization, we annotated additional 9,478 videos (vision+sound) which are not seen by the trained networks before. This new dataset consists of 44 categories from 6 major groups of concepts (i.e. urban, nature, work/home, music/entertainment, sports, and vehicles). It is annotated by Amazon Mechanical Turk workers. The frequency of categories depend on natural occurrences on the web, hence unbalanced.

**Vision vs. Sound Embeddings:** In order to show the semantic relevance of the features, we performed a two dimensional t-SNE [38] embedding and visualized our dataset in figure 4. The visual features are concatenated `fc7` features of the two VGG networks trained using ImageNet and Places2 datasets. We computed the visual features from uniformly selected 4 frames for each video and computed the mean feature as the final visual representation. The sound features are the `conv7` features extracted using SoundNet trained with VGG supervision. This visualizations suggests that sound features alone also contain considerable amount of semantic information.

**Object and Scene Classification:** We also performed a quantitative comparison between sound features and visual features. We used $60\%$ of our dataset for training and the rest for the testing. The chance level of the task is $2.2\%$ and choosing always the most common category (i.e. music performance) yields $14\%$ accuracy. Similar to acoustic scene classification methods, we trained a multi-class SVM over both sound and visual features individually and then jointly. The results are displayed in Table 7. Visual features alone obtained an accuracy of $49.4\%$. The SoundNet features obtained $32.4\%$ accuracy. This suggests that even though sound is not as informative as vision, it still contains considerable amount of discriminative information. Furthermore, sound and vision together resulted in a modest improvement of $2\%$ over vision only models.

### 4.4 Visualizations

In order to have a better insight on what network learned, we visualize its representation. Figure 5 displays the first 16 convolutional filters applied to the raw input audio. The learned filters are diverse, including low and high frequencies, wavelet-like patterns, increasing and decreasing amplitude filters. We also visualize some of the hidden units in the last hidden layer (`conv7`) of our sound representation

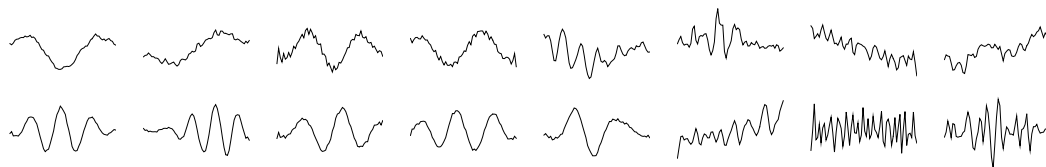

Figure 5: **Learned filters in conv1:** We visualize the filters for raw audio in the first layer of the deep convolutional network.

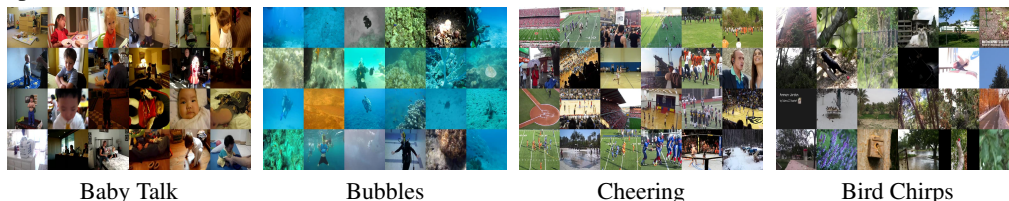

Baby Talk        Bubbles        Cheering        Bird Chirps

Figure 6: **What emerges in sound hidden units?** We visualize some of the hidden units in the last hidden layer of our sound representation by finding inputs that maximally activate a hidden unit. Above, we illustrate what these units capture by showing the corresponding video frames. No vision is used in this experiment; we only show frames for visualization purposes only.

by finding inputs that maximally activate a hidden unit. These visualization are displayed on Figure 6. Note that visual frames are not used during computation of activations; they are only included in the figure for visualization purposes.

## 5   Conclusion

We propose to train deep sound networks (SoundNet) by transferring knowledge from established vision networks and large amounts of unlabeled video. The synchronous nature of videos (sound + vision) allow us to perform such a transfer which resulted in semantically rich audio representations for natural sounds. Our results show that transfer with unlabeled video is a powerful paradigm for learning sound representations. All of our experiments suggest that one may obtain better performance simply by downloading more videos, creating deeper networks, and leveraging richer vision models.

**Acknowledgements:** We thank MIT TIG, especially Garrett Wollman, for helping store 26 TB of video. We are grateful for the GPUs donated by NVidia. This work was supported by NSF grant #1524817 to AT and the Google PhD fellowship to CV.

## Footnotes

\*contributed equally

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
