[Reviews · NeurIPS 2016]

Reviewer 1

Summary

This paper describes a method to learn audio representations from unlabeled audio+video data. The key idea is to regress a pre-trained visual recognition system's output from acoustic inputs. The authors collected a large corpus of training data, and evaluated the idea on standard benchmark datasets. A brief qualitative analysis of the learned features is also provided.

Qualitative Assessment

I enjoyed this paper. The idea is simple, but effective, and the results are impressive. The presentation was generally clear, but could be improved with a bit more detail in parts. Since the submitted draft is a bit short of the page limit, there should be no problem in filling in the following details: - The network illustrated in figure 1 could be explained in more detail. It's not clear from the figure what the filter sizes or pooling dimensions are. This would be helpful for interpreting figure 3, which is otherwise unlabeled. - The loss function (equation 1) is not fully specified. Are the teaching signals outputs g_k(y_i) interpreted as a categorical (multi-class) distribution, or a collection of independent binary (multi-label) distributions? Either could make sense in this context. - Tables 1 and 2 provide aggregated scores, but if there's space, it might be instructive to see some kind of error analysis or confusion matrices. Are there certain types of classes for which the proposed system performs well or poorly, or are mistakes spread evenly? I would expect this to depend on the relative distributions of concepts between, say, DCASE and the training data. If so, some brief discussion of this effect would be informative. - Line 154: "[Same] data augmentation is also applied during testing..." please provide detail about this. It sounds important, but is not possible to reproduce from what is written here. Typos: - line 58: "loosing" -> "losing" - line 154: "Same" -> "Some"

Confidence in this Review

3-Expert (read the paper in detail, know the area, quite certain of my opinion)


Reviewer 2

Summary

The paper presents a scheme for learning semantic representations of natural sound from videos not labeled with acoustic categories by transferring information between visual and acoustic representations. The authors use pre trained vision networks to supervise the learning of a convolutional network operating on sound. The representations learned by the sound network are then found to be useful for discriminating between semantic acoustic categories. Experiments demonstrating promising results are presented.

Qualitative Assessment

The technical novelty of the paper is minor. The approach presented in the paper is a straightforward combination of existing approaches and there doesn't appear to be any major technical innovations. Detailed comments: 1) The authors could also be more thorough with their experiments. In particular, a) It would be interesting to see how a purely unsupervised approach compares against the student-teacher model employed here. For example, it is easy to train a stacked convolutional auto encoder on the raw sound waves. Analogous features could then be extracted from the auto encoder and fed into the authors' sum pipeline. b) The authors use both a place and an object visual network to supervise the sound network. I would like to see a more careful experiment controlling for the effect of these networks. How does the performance vary if only one of the networks is used for supervision? For example, does knowing that a car and a bus are in the video boost performance over just knowing that an urban scene is captured in the video. 2) I also find it curious that in Equation 1, the authors minimize the reverse KL instead of minimizing the KL between the “true” distribution g and the estimate f, KL (g || f). What was the rationale behind minimizing the reverse KL. While "unsupervised" learning of natural sound representations is interesting and the authors present promising results, given the lack of technical contributions, the paper may be better suited for a more topical conference.

Confidence in this Review

3-Expert (read the paper in detail, know the area, quite certain of my opinion)


Reviewer 3

Summary

The authors start with an existing trained ImageNet vision network, run it overunlabeled videos, and try to predict the vision network outputs using audio alone. The authors then use various intermediate layers of the audio network as a feature on benchmark audio event classification tasks.

Qualitative Assessment

Full disclosure: I am involved in some *very* similar work to this, so this is a bit hard to review --- I feel slightly scooped. Hats off to the authors of this paper. The central idea of the paper, to leverage the power of highly trained vision networks to build audio classifiers, is strong. I've heard the same idea from multiple places, but this is the first time I've seen it fully executed, and ideas are cheap and execution at this scale is not trivial, so in that sense it is novel. The choice to use a fully convolutional net to handle arbitrary length audio sequences seems reasonable to me. I note that the authors chose to minimize the KL divergence against the output of the vision net [Eq 1]. Other reasonable choices would be L2 or cosine against some embedding layer of the vision system. It would be interesting to see some discussion or comparison here, but I think the choice made is fine. The biggest problem with all of this [which is not the authors' fault] is that it is impossible to get excited about any of the existing benchmarks. They are all quite small and not well explored. The idea that we could get gains by leveraging unlabeled video is reasonable and promising and matches my experience, but I'm reviewing another paper for this conference where similar gains on ECE-50 came just from messing around with the network architecture [no outside data], so putting that together, I'd say these results are only "suggestive." Again, this is not the authors' fault --- it's not like there are better datasets lying around that they chose to ignore. It would've been interesting / fun to see a little more analysis of what kinds of sounds the network was especially accurate or inaccurate at. So, overall, a fine paper. A good idea, executed well. Minor comments: Line 154: "Same data augmentation procedure is also applied during testing." I'm not sure what the procedure is. Line 177: s/tranining/training/

Confidence in this Review

3-Expert (read the paper in detail, know the area, quite certain of my opinion)


Reviewer 4

Summary

This paper proposes to train a representation of natural sounds on a dataset of unlabeled video by leveraging (transfer-learning) the "knowledge" from a vision-based neural network that has been very well trained on a separate large dataset of labeled visual data. The paper shows that the obtained audio features can lead to state-of-the-art performance when used as input to a multi-class SVM on two audio scene classification benchmarks.

Qualitative Assessment

This is a simple but interesting idea, and the results are promising. There are several points that could be improved: 1) What is the motivation for the particular architecture that was chosen? No other architecture is investigated, and this choice seems rather ad hoc. There exist many other neural networks that are being used in audio, whether for automatic speech recognition or for source separation, and I have yet to see such a network architecture. Why not simply use a DNN or LSTM on a frame-wise representation of the signal, e.g., STFT magnitude or log-magnitude? It would have then been easy to synchronize the audio and the video. There are any number of possibilities for alternative architectures. Even within the realm of the proposed architecture, what is the motivation for having, e.g., a pool5 layer and not a pool6 layer? Table 2 does not answer such questions. 2) Presentation is okay but not always clear. Most importantly, the main transfer-learning procedure shown in Figure 1 is not well explained until section 5.3, and the fact that 5.3 is an explanation for that procedure is not even clear. There should be a dedicated section that explains how transfer-learning is performed in practice, in particular how the visual category distribution that is used in the KL divergence is defined. The experimental procedure for DCASE and ESC could also be made clearer: in particular, the expression "the mean score of all sound excerpts is used as the final score of a test recording" could be misleading (I first thought that the "score" referred to here was recognition accuracy, before realizing it was the SVM scores over categories). 3) I think it would be more insightful to show the conv5 or pool5 sound features instead of the conv7 ones. Arguably, the further in the network architecture, the more likely (or less surprising) it is that the sound features will have similarly discriminative characteristics to visual features, because that's how they have been trained. Figure 2 can serve as a confirmation that training went well, but showing similar results for conv5 or pool5 would be more interesting as those layers are the ones that work best on the sene identification tasks. Same remark for the visualization. 4) Certainly there are other insights to be shared and results to be shown: it's a shame to stop the paper at the beginning of page 7.

Confidence in this Review

2-Confident (read it all; understood it all reasonably well)


Reviewer 5

Summary

This paper propose to use pre-trained CNNs for visual recognition to be used as a teacher to the learn a student sound network on the 2M acoustic unlabelled videos taken from fliker to learn natural sound representations. The idea of transferring abstract knowledge learned from one modality to other in the student teacher network is interesting and looks appealing. Although the methodology used is not very novel it facilitate learning of much deeper sound-net. The convolution features learned are shown to give much better results then that of the hand crafted features thus the experimental results are encouraging. The presentation of the sound net architecture is clear and most of the design choices are justified. Over all a well presented paper with incremental novelty but with good results. The most crucial drawback of the paper is the absence of a reasonable baseline CNN which is trained on a large enough data-set without supervision. A simple soundnet based autoencoder as the unsupervised network should be trained on 2M training images as the baseline for acceptance of the paper. My recommendation is a weak reject but I will be happy to change my rating if in the rebuttal authors can substantiate the utility of the teacher student setup of figure 1 over convolutional Auto encoder.

Qualitative Assessment

The most crucial drawback of the paper is the absence of a reasonable baseline CNN which is trained on a large enough data-set. The largest labelled sound dataset used for evaluation is ESC-10/50 with 2000 sound recordings which is too small for training a CNN from scratch. In table 1a) all methods are said to be hand crafted features except that of [24] which is supervised and trained on 2000 sound recordings? I believe that the source if improvement over [24] could be (i) the difference in architecture: this can be simply ruled out by training sound-net on ESC as a replacement of [24] and show that the sound-net is too deep to be trained directly on such small datasets. (ii) The knowledge transferred from state of the art visual recognition networks (Alex-net/VGG) as claimed by the paper. (iii) Simply because of the fact that we have now a large unlabelled dataset. Given over 2M soundbites at our disposal, a very simple baseline is a traditional convolution autoencoder with tied weights similar to sound-net encoder. This encoder can be fine tuned on ECS and iHear separately for comparison. (A simple classifier on the autoencoder codes can also be used). This experiment is mandatory to verify weather the vision to audio knowledge transfer is useful. i.e how much improvement you get from merely using huge unlabelled data-set and quantify the contribution of visual training. Another very important question is that flicker is said to mined with keywords. Although, these keywords does not correspond to the classification labels in the test experiment why can't we learn a siamese network with these surrogate labels and then use finetuning? Other minor comments: - a brief description of these baselines is needed as the reader is bound to refer to line 47-48 even to know that out of so many baselines only [24] is a conv-net for which the architecture is not outlined in the paper). - As iHear seems to be more diverse and biggest labelled data-set, all the baseline (especially [24]) should be tested also on iHear. - Why other unsupervised and supervised deep learning methods apart from [24] is not compared against?

Confidence in this Review

2-Confident (read it all; understood it all reasonably well)


Reviewer 6

Summary

This paper proposed a student-teacher training procedure to learn acoustic representations from large collections of unlabeled videos, which transfers discriminative visual knowledge from well established visual models into sound domain using unlabeled video as a bridge.

Qualitative Assessment

This paper proposed an interesting way to construct the pseudo-label to learn natural sound representations from unlabeled video, which has delivered promising results on classification problems. I think this is one of a few which tried to directly transfer the "label"s using an existing classifier to learn the representations. To me, it is a bit surprising since there seem not have many category overlaps among the ImageNet/Place dataset and the natural world videos. Some discussions of why this works and possible experiment explorations would be nice. As a baseline, it would also be good to show the performance of the SoundNet trained using labels randomly generated from 1000/401 categories randomly. How would the selection of frames in the image-based ConvNet impact the final performance? It would have been interesting to test the robustness of the proposed method to different ways of selecting frames (randomly / most diverse / representative), and different numbers of chosen frames. Also, the video-based ConvNet ([1] for example) may be a better fit to eliminate this burden. It is easy to follow the paper, but I think the section 4.1 can be expanded to elaborate more on how to handle variable-length input since the recording length used in the experiment section is different from that used in the training stage. [1] A. Karpathy et al. Large-scale video classification with convolutional neural networks. CVPR 2014.

Confidence in this Review

2-Confident (read it all; understood it all reasonably well)